# Shadow Detection for Ultrasound Images Using Unlabeled Data and Synthetic Shadows

**Suguru Yasutomi**[1,2]                                           YASUTOMI.SUGURU@FUJITSU.COM
**Tatsuya Arakaki**[3]                                              ARAKAKIT@MED.SHOWA-U.AC.JP
**Ryuji Hamamoto**[4,5]                                            RHAMAMOT@NCC.GO.JP

[1] *Artificial Intelligence Laboratory, Fujitsu Laboratories Ltd.*

[2] *RIKEN AIP-Fujitsu Collaboration Center, RIKEN*

[3] *Department of Obstetrics and Gynecology, Showa University School of Medicine*

[4] *Cancer Translational Research Team, AIP Center, RIKEN*

[5] *Division of Molecular Modification and Cancer Biology, National Cancer Center Research Institute*

## 1. Introduction

Medical ultrasound is a popular image diagnosing technique. The advantage of ultrasound imaging is low introduction cost and high temporal resolution. On the other hand, its spatial resolution tends to be low, and this leads clinicians to overlook small lesions. To alleviate this, there have been many attempts to support diagnosing by image recognition (Noble and Boukerroui, 2006; Cheng et al., 2010). However, ultrasound images often suffer from shadows caused mainly by bones. The shadows prevent not only diagnosing accurately from clinicians but also working properly from the image recognition methods. Detecting such shadows can be the first step to deal with them. Once shadows detected, we can screen data for image recognition as preprocessing, alert clinicians to low-quality images, and so forth. Shadow detection methods had been performed using traditional image processing (Hellier et al., 2010; Karamalis et al., 2012). Recently, deep learning based methods have been proposed (Meng et al., 2018a,b). The traditional methods rely on domain-specific knowledge, and thus it is costly to apply to multiple different domains. The deep learning based methods learn optimal feature extraction automatically from training data, but they need segmentation labels which are expensive.

In this paper, we propose a novel shadow detection method that can be learned using only unlabeled data by utilizing the feature extraction capability of deep learning and relatively coarse domain-specific knowledge. Namely, our method is based on a restricted auto-encoding (Vincent et al., 2010) structure and synthetic shadows. We construct the structure that separates input images into shadows and other contents and then combines them to reconstruct input images. To encourage the network to separate the input into these images, it is learned to predict synthetic shadows that are injected to the input beforehand. We show that our method can detect shadows effectively by experiments on ultrasound images of fetal heart diagnosis.

## 2. Proposed method

### 2.1. Network Structure

Figure 1 shows the overview of the proposed method. The network archtecture is based on auto-encoders, but it consists of 1) single encoder $E$ that extracts feature $z$, 2) shadow decoder $D_s$ that predicts shadow images $\hat{x}_s$, 3) content decoder $D_c$ that predicts content without shadows $\hat{x}_c$, and 4) generation of reconstruction images $\hat{x}$ by element-wise product of $\hat{x}_s$ and $\hat{x}_s$. The encoder $E$ is

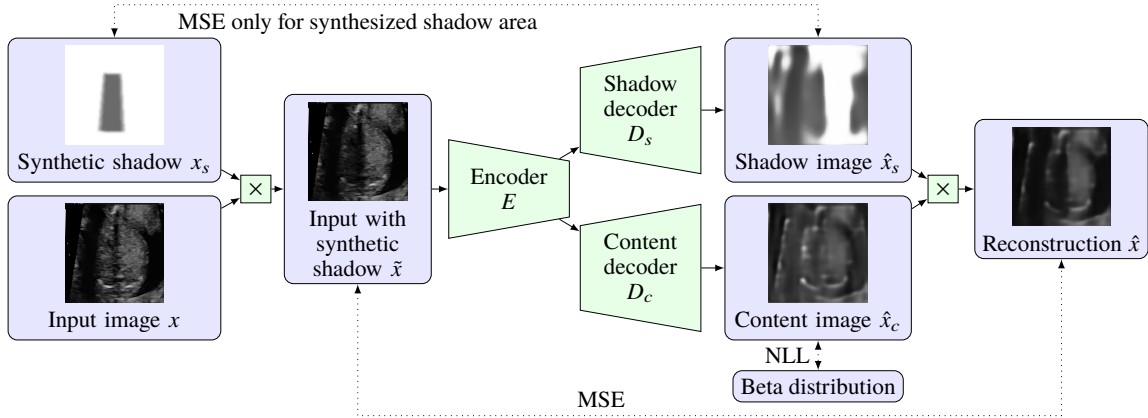

Figure 1: Overview of the proposed method. Solid arrows and dotted arrows represent data flow and calculation of losses, respectively.

a ConvNet, and the decoders $D_s$ and $D_c$ are DeconvNets. The features and images are defined as follows:

$$z = E(x), \ \hat{x}_s = \mathrm{sigmoid}(D_s(z)), \ \hat{x}_c = \mathrm{sigmoid}(D_c(z)), \ \hat{x} = \hat{x}_s \circ \hat{x}_c,$$

where $x \in [0,1]^{W \times H}$ is the input image and $\circ$ denotes Hadamard product. The loss function for the auto-encoding structure is given as mean squared error (MSE), i.e. $l_{\mathrm{AE}}(x, \hat{x}) = \frac{1}{WH} \sum_{ij} (\hat{x}_{ij} - x_{ij})^2$.

### 2.2. Synthetic Shadow Injection and Prediction

The normal learning process for auto-encoders does not lead $\hat{x}_s$ and $\hat{x}_c$ to contain only shadows and only contents, respectively. To make $D_s$ output only shadows, we inject synthetic shadows $x_s$ to input images and lead $D_s$ to predict them. As we know which kind of shadows may appear depending on the type of probes, it is relatively easy to inject plausible synthetic shadows. In this work, we focus on convex probes and generate shadows as random annular sectors in a rule-based manner. The injection of the synthetic shadows $x_s \in [0,1]^{W \times H}$ can be written as $\tilde{x} = x \circ x_s$, and we replace the input to the network from $x$ to $\tilde{x}$. Regarding that the background of $x_s$ is 1 and the synthetic shadows are expressed as a range of $[0,1)$, we define the loss function for shadow prediction as

$$l_s(x_s, \hat{x}_s) = \frac{1}{WH} \sum_{ij} \mathbf{1}[\hat{x}_{s_{ij}} \neq 1](\hat{x}_{s_{ij}} - x_{s_{ij}})^2.$$

It evaluates the correctness of predicted shadows w.r.t. the area that the synthetic shadows exist because we do not know whether shadows exist on other area.

### 2.3. Loss function

Besides loss functions mentioned above, we use two additional functions. One is regularization for predicted shadows $\hat{x}_s$ that prevent them from getting too dark: $l_{\mathrm{sreg}}(\hat{s}) = \frac{1}{WH} \sum_{ij} |1 - \hat{x}_{s_{ij}}|$. The other is to restrict the predicted content $\hat{x}_c$ to distributed under some beta distribution (Bishop, 2006): $l_c(\hat{x}_c) = -\sum_{ij} \ln \left[ p_{\mathrm{beta}}(\hat{x}_{c_{ij}} | \alpha, \beta) \right]$, i.e. negative log likelihood (NLL) under the distribution. The resulting loss function is given as

$$l = \lambda_{\mathrm{AE}} l_{\mathrm{AE}}(\tilde{x}, \hat{x}) + \lambda_s l_s(x_s, \hat{x}_s) + \lambda_{\mathrm{sreg}} l_{\mathrm{sreg}}(\hat{x}_s) + \lambda_c l_c(\hat{x}_c),$$

where $\lambda_{\mathrm{recon}}, \lambda_{\mathrm{shadow}}, \lambda_{\mathrm{sreg}}$, and $\lambda_{\mathrm{content}}$ are weights for the losses.

Table 1: Experimental results. Each score is average over images.

| Method | IoU | DICE |
|---|---|---|
| Trivial thresholding | 0.229 (±0.105) | 0.361 (±0.140) |
| SegNet | 0.338 (±0.150) | 0.486 (±0.179) |
| Proposed method | **0.340** (±0.132) | **0.492** (±0.161) |

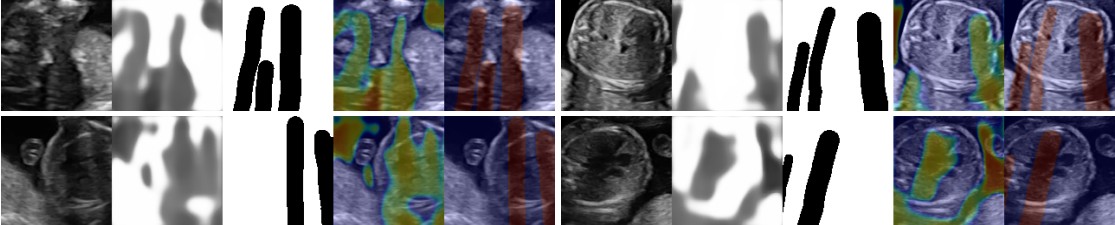

Figure 2: Example of shadow detection. From left to right; input image, shadow prediction, ground truth, shadow prediction overlayed to input, and ground truth overlayed to input.

## 3. Experiments

We evaluated the performance of the proposed method using ultrasound images of fetal heart diagnosis. Data for the experiments were acquired in Showa University Hospital, Showa University Toyosu Hospital, Showa University Fujigaoka Hospital, and Showa University Northern Yokohama Hospital. All the experiments were conducted in accordance with the ethical committee of each hospital. Dataset consisted of 107 videos of 107 women who are 18–34 weeks pregnant. The dataset was split into training data, validation data, and test data. Training data was 93 videos which were converted to 37378 images. The validation data was 61 images picked up from 7 videos. The test data was 52 images picked up from 7 videos. The validation data and the test data were pixel-level annotated by clinicians. We compared the performance of the proposed method with trivial image thresholding as a baseline, and with SegNet (Badrinarayanan et al., 2017) as a reference of simple deep learning approach. Because SegNet is a supervised method, we trained it with a part of the validation data. Five videos in the validation data were used as training data for SegNet, leaving two videos for validation. For all methods, the hyperparameters were selected using the validation data. Table 1 shows the results. It shows that our method fairly more effective than the trivial one. Moreover, the proposed method achieved slightly better performance than SegNet. This result indicates that our method works well in situations with small annotated data. Figure 2 shows examples of shadow detection with our method. They illustrate that our method can detect various shadows, although it tends to predict dark areas (e.g. amniotic fluid and cardiac cavity) as shadows.

## 4. Conclusion

In this paper, we proposed a shadow detection method for ultrasound images that can be learned by unlabeled data. By experimental results, the effectiveness of the method is shown. Since the method uses only unlabeled data, it can be easy to apply to multiple different domains (e.g. different machines and different organs). Evaluations in such situations would be future work.

## Acknowledgments

We would like to thank Akira Sakai, Masaaki Komatsu, Ryu Matsuoka, Reina Komatsu, Mayumi Tokunaka, Hidenori Machino, Kanto Shozu, Ai Dozen, Ken Asada, Syuzo Kaneko, and Akihiko Sekizawa for useful discussions and data acquisition.

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
