# OpenReview forum: "Shadow Detection for Ultrasound Images Using Unlabeled Data and Synthetic Shadows"
_MIDL.io/2019/Conference/Abstract — MIDL Abstract 2019_

### Official Review · AnonReviewer1 · 2019-04-28
**A nice approach to remove shadow artefacts from ultrasound images.**

**Rating:** 3
**Confidence:** 2

**Review:**

The abstract introduces an intuitive approach to decompose ultrasound images into the artefact and image components. The idea of using a realistic artefact simulation to train the model to separate the shadow artefacts purely achieves a reasonable detection accuracy of artefacts. The most useful aspect of this work is the ability of the proposed model to not only detect the artefacts but also to correct for them, which would not be possible with standard segmentation-based approaches.

---

### Official Review · AnonReviewer2 · 2019-05-02
**Interesting Work**

**Rating:** 3
**Confidence:** 2

**Review:**

The authors propose an automatic shadow detection method for Ultrasound images that does not require manually labeled training data. This is certainly useful for improving the quality of ultrasound images.

The method is based on an encoder-decoder architecture and theoretically sound. Synthetic shadows are used to train the network hence eliminating the need for manual labels.

Results are demonstrated on a dataset of 107 videos of 107 patients, 93 of which are used as training, 7 as validation and 7 as test. In order to make a quantitative comparison, manual labeling was done on the validation and test sets. Due to the difficulty of this process, the number of videos used for evaluation is reasonable.

The method performs significantly better than trivial thresholding.

The improvement over SegNet is very minimal but SegNet requires manually labeled data and is hence trained only on 5 of the videos from the validation set so this comparison shows that the proposed method achieves similar performance to a baseline supervised semantic segmentation method when a small amount of labeled data is available.

Adding comparison to another recent method that does not require labeled data would make experimental evaluation more convincing.

The English in the paper could use some improvement. There are a few grammar issues.

Quality: 3/5
Clarity: 4/5
Originality: 3/5
Significance: 3/5

---

### Decision · Program_Chairs · 2019-05-06
**Acceptance Decision**

Accept